# Tri- and Pentacyclic Azaphenothiazine as Pro-Apoptotic Agents in Lung Carcinoma with a Protective Potential to Healthy Cell Lines

**DOI:** 10.3390/molecules27165255

**Published:** 2022-08-17

**Authors:** Magdalena Skonieczna, Anna Kasprzycka, Małgorzata Jeleń, Beata Morak-Młodawska

**Affiliations:** 1Biotechnology Centre, Silesian University of Technology, 44-100 Gliwice, Poland; 2Department of Systems Biology and Engineering, Faculty of Automatic Control, Electronics and Computer Science, Silesian University of Technology, 44-100 Gliwice, Poland; 3Department of Organic Chemistry, Bioorganic Chemistry and Biotechnology, Faculty of Chemistry, Silesian University of Technology, 44-100 Gliwice, Poland; 4Department of Organic Chemistry, Faculty of Pharmaceutical Sciences in Sosnowiec, The Medical University of Silesia in Katowice, Jagiellońska 4, 41-200 Sosnowiec, Poland

**Keywords:** dipyridothiazine, diquinothiazine, cancer cell lines A549 and H1299, anticancer activity, apoptosis, long-term live cell observations, BEAS-2B and healthy NHDF cell lines

## Abstract

The phenothiazine derivatives, tricyclic 10*H*-3,6-diazaphenothiazine (**DPT-1**) and pentacyclic 7-(3′-dimethylaminopropyl)diquinothiazine (**DPT-2**), have recently been shown to exhibit promising anticancer activities in vitro. In this report, we demonstrated that **DPT-1** and **DPT-2** could be pro-apoptotic agents in lung carcinoma, the human lung carcinoma A549 and non-small lung carcinoma H1299, in the range of IC_50_ = 1.52–12.89 µM, with a protective potential to healthy cell lines BEAS-2B and NHDF. The compounds showed higher activity in the range of the tested concentrations and low cytotoxicity in relation to normal healthy cells than doxorubicin, used as the reference drug. The cytostatic potential of **DPT-1** and **DPT-2** was demonstrated with the use of MTT assay. Cell cycle analysis via flow cytometry using Annexin-V assay showed the pro-apoptotic and pro-necrotic role of the studied diazaphenothiazines in the cell cycle. **DPT-1** and **DPT-2** initiated a biological response in the investigated cancer models with a different mechanism and at a different rate. Based on these findings, it can be concluded that **DPT-1** and **DPT-2** have potential as chemotherapeutic agents.

## 1. Introduction

Worldwide, lung cancer has been the most common diagnosed carcinoma for the last several decades. It is the world’s leading cause of cancer death [1,2]. This is due to the fact that it is completely asymptomatic in the initial stage and usually detected in the advanced stages [3]. Globally, lung cancer cases and deaths are rising. In 2018, GLOBOCAN indicated 2.09 million new cases (11.6% of total cancer cases) and 1.76 million deaths (18.4% of total cancer deaths), higher than 2012 reported rates making it the most frequent cancer and cause of cancer death in men and women combined [4,5,6,7]. Tobacco smoking remains the biggest risk factor for lung cancer. However, nontobacco risk factors such as environmental and occupational exposures, chronic lung disease, and lifestyle factors contribute to lung cancer risk too [1]. Therapies for lung cancer include surgical removal, radiotherapy, and chemotherapy. Chemotherapy was reported to be the most efficient treatment, although it is commonly associated with side effects on normal cells [8]. Therefore, it is imperative to discover and develop new, more potent anticancer agents with better selectivity and reduced side effects.

The heterocyclic ring system plays an important role in the development of novel scaffolds with improved pharmaceutical properties in anticancer research [9]. Tricyclic phenothiazines are an important class of heterocycles showing important biological and chemical properties. For many years, this class of organic compounds has been recognized as neuroleptic, antihistaminic, antitussive, and antiemetic drugs [10,11]. Novel derivatives of phenothiazines have been obtained via modification of the parent phenothiazine via the introduction of a new substituent at the thiazine nitrogen atom or via the substitution of one or two benzene rings with homoaromatic and heteroaromatic rings. As a result, azaphenothiazines were obtained containing a pyridine and quinoline ring (or rings) in their structure [12,13]. Modified phenothiazines have exhibited promising biological activities such as anticancer, antibacterial, and potential treatment in Alzheimer’s and Creutzfeldt-Jakob diseases. These numerous scientific reports are the subject of many reviews that have appeared in world literature [14,15,16]. Additionally, azaphenothiazines (dipyridothiazines and diquinothiazines) are promising heterocycles with anticancer, immunosuppressive, and antioxidant properties [17,18,19].

Among the anticancer active diazaphenothiazine (**DPT**) series, two derivatives deserve special attention: tricyclic 10*H*-3,6-diazaphenothiazine (dipyrido [2,3-b;4′,3′-e][1,4]thiazines) (**DPT-1**, with two pyridine rings) and pentacyclic 7-(3′-dimethylaminopropyl)diquinothiazine (**DPT-2**, with two quinoline rings) (Figure 1). From a chemical perspective, **DPT-1** was obtained efficiently via a Smiles rearrangement reaction using a 3-amino-3′-nitro-2,4′-dipyridinyl sulfide [20]. In contrast, **DPT-2** was obtained in a multi-step synthesis using 2,2′-dichloro-3,3′-diquinolinyl disulfide and diquinodithiin [21]. 10*H*-3,6-diazaphenothiazine (**DPT-1**) exhibited an extremely strong action against the glioblastoma SNB-19, melanoma C-32, and breast cancer MCF-7 cell lines with IC_50_ values of 0.46 and 0.72 μg/mL and non-toxic action against the normal fibroblast HFF-1 cell line [20]. This compound induced apoptosis through upregulation of pro-apoptotic genes such as Bax, p53, and CDKN1A (p21) and downregulation anti-apoptotic genes such as Bcl-2 and H3 (a histone indicator for proliferation of cellular DNA) [20]. Anticancer action of **DPT-1** was also studied on A2780 ovarian cancer cells via an investigation on cytotoxicity profiles, the mechanism of apoptosis, and cell invasion. This compound induced a dose-dependent inhibition on A2780 cancer cells (IC_50_ = 0.62 μM), with significantly less cytotoxicity towards normal kidney HEK293 cells and normal heart H9C2 cells. This compound induced the generation of reactive oxygen species (ROS) and the polarization of mitochondrial membrane potential (ΔΨm). It is connected with inducting cell death through oxidative damage. This compound elicited an upregulation of caspase-6, -3, and -7, which are actively involved in the formation of cell shrinkage, chromatin condensation, and the fragmentation of DNA. Additionally, the activation of caspase-3 brought about increased enzymatic activity of DFFA (DNA fragmentation factor-α). **DPT-1** induced apoptosis via intrinsic (mitochondria-dependent) and extrinsic (cell death receptor-dependent) pathways [22].

Pentacyclic 7-(3′-dimethylaminopropyl)diquinothiazine (**DPT-2**) exhibited strong antiproliferative activity in vitro against glioblastoma SNB-19, colorectal carcinoma Caco-2, breast cancer MDA-MB-231, and lung cancer A549 cell lines (activity in the range of IC_50_ = 0.3–3.44 μM). **DPT-2** characterized low cytotoxicity against normal human dermal fibroblasts NHDF. Significant anti-proliferative activity of **DPT-2** against human cancer cell lines and low cytotoxicity prompted a careful study of its activity against lung cancer tumor cell lines [21].

Based on the promising results above, the aim of the current study is to investigate anticancer activities and the detailed apoptosis pathway induced by **DPT-1** and **DPT-2** towards two lung cancer cell lines: the human lung carcinoma (A549), non-small lung carcinoma (H1299), and human non-tumorigenic lung epithelial cell line (BEAS-2B), with comparison to normal human dermal fibroblasts (NHDF). Regarding **DPT-1**, these studies are completely innovative, while in the case of **DPT-2**, they are an extension of earlier preliminary studies.

## 2. Results and Discussion

### 2.1. Anticancer Activity

In the first stage of the research, the antitumor potential of both compounds was determined, as shown in Table 1, Figure 2, and Figure 3 with comparison to doxorubicin as a reference drug. Cellular viability was estimated after 72 h of incubation with tested compounds, and for IC_50_ the dose-effect calculations were performed [23,24,25]. A standard anticancer drug, such as doxorubicin, was simultaneously used at the same concentrations range: 100, 50, 25, 12.5, 6.25, 3.15, 1.56 µM. Finally, the cell line’s sensitivities were established, whereby the tested phenothiazine **DPT-1**, according to the IC_50_ values, was the most active towards the cancer A549 cell line (Table 1).

In comparison to the doxorubicin, which is the most active against human non-small cell lung carcinoma cell line, H1299 (IC_50_ = 7.749 ± 0.004 µM), selectiveness against both cancer cell lines, A549 and H1299, is also preset for **DPT-1** and **DPT-2**. **DPT-1** is the most active against the A549 cell line (IC_50_ = 1.526 ± 0.004 µM) and then against H1299 (IC_50_ = 2.515 ± 0.005 µM); normal fibroblasts NHDF are sensitive at a similar level (IC_50_ = 2.246 ± 0.01 µM) as the human bronchial epithelial cells, BEAS-2B (IC_50_ = 2.4679 ± 0.01µM). Structural changes of phenothiazines in **DPT-2** decreased its activities a little bit against A549 (IC_50_ = 3.447 ± 0.054 µM) and H1299 (IC_50_ = 12.895 ± 0.013 µM) cancer cell lines. However, the cytotoxicity of the compound **DPT-2** against normal cells NHDF or BEAS-2B is not observed at the same doses (IC_50_ = 18.77 ± 0.038 µM and IC_50_ = 11.2648 ± 0.038 µM, respectively). These findings are promising for the application of modified phenothiazines **DPT-1** and **DPT-2** in combined chemotherapies, e.g., with doxorubicin exclusion for more sensitive patients, because of strong anticancer effects decreasing the viability of cancer cell lines A549 and H1299 and at the same time a lower toxicity observed in normal NHDF and BEAS-2B cells (Table 1). That means the lowest doses of **DPT-1** or **DPT-2** are not as toxic in healthy cell lines. When doxorubicin is used against cancer cells, the side effect is observable in the healthy neighborhood, e.g., in epithelia, for BEAS-2B the cytotoxicity of this drug is very high (IC_50_ = 0.0651 ± 0.001 µM), which excludes it from therapies. Phenothiazines are good candidates for in vitro testing, especially on 3D cultures where a mix of sensitive healthy cell lines are used together with chemoresistant cancer cell lines. A low lethal effect is observed in NHDF or BEAS-2B cells at doses of IC_50_ against A549 or H1299 cells, hence, the selectivity against cancers is confirmed (Table 1).

Viability results, followed by MTT 72 h assay, showed better anticancer potential against H1299 cells than that observed for doxorubicin (positive control on Figure 2), especially for **DPT-1**. Both tested compounds displayed typical dose-effect activities at low doses, between 1.56 and 6.25 µM, which confirms previous findings for bioactive molecules, such as phenothiazines.

The A549 cell line used seems to be more resistant to the tested compounds, **DPT-1** and **DPT-2**; additionally, the lowest sensitivity was observed against doxorubicin (Figure 3). All these findings suggest the resistance of the adenocarcinoma human alveolar basal epithelial cells, as the A549-DOX-res cell line. Although the results look similar at dosing protocols for doxorubicin, the role of the phenothiazines **DPT-1** and **DPT-2**, as an alternative for the doxorubicin-resistant cancer cell lines anticancer therapies, is still promising.

### 2.2. Cytostatic Potential of Modified Phenothiazines

In the second stage of the research, the role of the phenothiazines **DPT-1** and **DPT-2** in the cancer cell lines A549 and H1299 as potentially cytostatic drugs was investigated in order to assess the antitumor activity (Figure 4).

A lowered viability, followed by the MTT assay, could be a perfect indicator of the anticancer potential of drugs. However, this procedure measures the mitochondrial activity of the enzymatic complex [24,25]. The real cytotoxicity against the antiproliferative action of drugs could be distinguished using the next step procedure, a cytometric cell cycle distribution, compared to the untreated controls [26,27,28]. For that reason, the cell cycle was measured, followed by 72 h of incubation with **DPT-1** and **DPT-2**. High doses of phenothiazines (50 µM) arrested the cell cycle at different phases, dependent on the cancer cell line. On typical histograms of cellular DNA content, the S and G2/M arrest is visible after 72 h of treatment with **DPT-1** and **DPT-2** in the A549 cell line (Figure 5A and Figure 6A). The phenomenon of sub-G1 phase reduction (lowered necrotic/apoptotic and general dead cells fraction), in comparison to the untreated control, could be explained by the cytostatic potential of phenothiazines. Their mode of action is rather antiproliferative than cytotoxic—the cells skipped a cell cycle, and the number of cells in comparison to the untreated control decreased. The MTT estimations of cells viability, calculated from the lowered absorbance, resulted from the antiproliferative and cytostatic role of **DPT-1** and **DPT-2** in the A549 cell line for S and G2/M cell cycle phases.

A different mode of action is presented in the H1299 cell line; the cell cycle is stopped in the G0/G1-phase after **DPT-1** incubation (50 µM), and a strong cytotoxic effect is presented after **DPT-2** addition, where the sub-G1 fraction (necrotic/apoptotic and dead cells fraction) as well as the G2/M fraction increased (Figure 5B and Figure 6B).

The high fraction of sub-G1 in controls of A549 and H1299 cells is connected with the regular proliferation during 72 h assay. At that time, the contact inhibition occurred and control cells started dying with the physiological process of apoptosis (Figure 4). Calculations, followed by cytometric cell cycle measurements, allowed for the inclusion of the background and the visualization of the proper cell cycle distribution (Figure 5) [26,27,28].

### 2.3. Pro-Apoptotic and Pro-Necrotic Role of Phenothazines

Followed by Annexin-V apoptosis assay after 72 h of incubation with **DPT-1** and **DPT-2** at a dose of 50 µM, the type of cellular death is possible to determine (Figure 6 and Figure 7). A prolonged treatment of cancer cell lines A549 and H1299 showed a pro-apoptotic activity of **DPT-1** in both cell lines, whereas **DPT-2** induced apoptosis, as well as necrosis (Figure 6A,B). After 72 h of incubation of cells in both controls, some of the early and late apoptotic cells were observable, without necrotic ones. Typical dot plots from cytometric measurements are presented in Figure 6.

**Figure 6 molecules-27-05255-f006:**
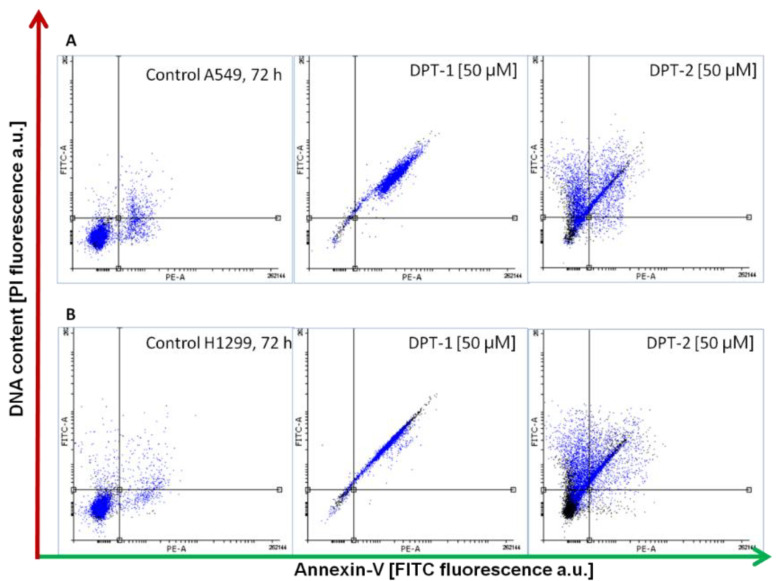
Typical dot plot followed by Annexin-V apoptosis assay and iodium propide (PI) staining. Cellular death in control and **DPT-1**- and **DPT-2**-treated A549 (**A**) and H1299 cells (**B**) treated for 72 h at a dose of 50 µM, followed by flow cytometry measurements (normal cells: Annexin-VI_−_/PI_−_; early apoptosis: Annexin-V_+_/PI_−_; late apoptosis: Annexin-V_+_/PI_+_; necrosis: Annexin-V_−_/PI_+_).

The dose of 50 µM, almost ten times higher than the calculated IC_50_ = 5.289 ± 0.004 µM in the A549 cancer cell line for doxorubicin, was used for induction of cellular death. **DPT-1** induced an apoptosis pathway after 72 h of treatment, which is present as a late apoptotic fraction (Annexin-V_+_/PI_+_). Both cancer cell line responded with almost 90% of cells apoptotic (Figure 7A,B). More toxic and lethal effects after a long-term exposure induced by **DPT-2** was observed, with circa 30 % of the tested population necrotic (Annexin-V_−_/PI_+_) (Figure 7A,B).

Much higher doses, than IC_50_ for the A549 cancer cell line (Table 1), for **DPT-1** and **DPT-2** (IC_50_ = 1.526 ± 0.004 µM and IC_50_ =3.447 ± 0.054 µM, respectively) showed the strong anticancer potential of modified phenothiazines, with both pro-apoptotic and pro-necrotic activities (Figure 8A,B). Considering the 72 h particular dose-dependent effects, at the lower doses, at a range of 0.78–12.5 µM of **DPT-1**, the necrosis is visible (Figure 8A). The higher doses, at a range of 25–100 µM of **DPT-1**, induced cellular death via an apoptosis pathway (Figure 8A). **DPT-2** at the same doses induced early apoptosis at a low range of 0.78–12.5 µM, whereas higher doses at a range of 25–100 µM induced toxic, necrotic death (Figure 8B). In resumption, that kind of sudden cellular death is desired in cancer cell lines; however, in the surrounding tissues it could also be pro-inflammatory [28,29].

### 2.4. Microscopic Long-Term Live Observations

The long-hour microscopic observations [30,31] shown in Figure 9 and Figure 10 confirm all previous observations.

Using both analogs of phenothiazines **DPT-1** and **DPT-2**, at the highest doses of 100 µM for 72 h of exposure, typical morphological changes for cellular death could be observed in lung epithelial cells BEAS-2B (Figure 9) or in lung carcinoma A549 cells (Figure 10). The cells stayed mostly unattached and floating with some shrunken, which indicates a late stage of apoptosis and necrosis. Using cancer cell lines A549 and H1299, the cellular apoptotic and necrotic death was confirmed, followed by cell cycle cytometric analysis for the sub-G1 fraction (please compare Figure 4 and Figure 5). Typical images taken from the BEAS-2B (Figure 9) or A549 cells (Figure 10) at a concentration of 50 µM resulted in a survival fraction at a level of 5–30% in comparison to the untreated controls, which was similar for **DPT-1** and **DPT-2** in A549 and H1299 cells, respectively (Figure 2 and 3). Such observations additionally explained the results of Annexin-V apoptosis assays, where some of the cells, mainly carcinomas, presented an early apoptosis fraction (Figure 6, Figure 7 and Figure 8). Lower doses, 25 µM, showed selectivity of phenothiazines, with more neutral impact on the healthy epithelial BEAS-2B cells (Figure 9) and slight cytotoxicity against lung carcinoma A549 (Figure 10), mostly for **DPT-1**.

### 2.5. ADME Analysis and Target Prediction of **DPT-1** and **DPT-2**

The tested compounds, **DPT-1** and **DPT-2**, were subjected to preliminary in silico analyses of pharmacokinetic parameters, an ADMET profile with bioavailability analysis, and prediction of biological targets using Web platform SwissADME [32] and Way2Drug [33] (Table 2).

These compounds show no significant differences in molecular descriptors and ADME parameters (Table 3 and Table 4). Basically, they differ in lipophilicity and molecular mass, which will undoubtedly contribute to the achievement of the molecular target. All tested derivatives meet the requirements of Lipinski’s rule of five and Ghose’s and Veber’s rules which point out that they could become drugs with the ability to be used as orally active drugs (Table 2). The tested compounds are characterized by positive passive human gastrointestinal absorption (GI), the ability to penetrate the blood-brain barrier (BBB), the ability of the permeability glycoprotein (Pgp), and the interaction with cytochromes P450 (major isoforms (CYP1A2, CYP2C9, CYP2D6)) (Table 3).

Using the Internet platform Way2Drug [33] and the PASS (*Prediction of Activity Spectra for Substance*) application available in it, the probability of interaction with the molecular targets of the tested compounds **DPT-1** and **DPT-2** was determined in order to confirm the validity of the research.

The obtained results (Table 4) confirmed the high probability of the compounds having an influence on the structure of histones and the tumor necrosis factor and an influence on angiogenesis. Additionally, possible anti-inflammatory and immunomodulatory effects were indicated respectively for **DPT-1** and **DPT-2**.

## 3. Materials and Methods

### 3.1. Chemicals

Tricyclic 10*H*-3,6-diazaphenothiazine (**DPT-1**, with two pyridine rings) and pentacyclic 7-(3′-dimethylaminopropyl)diquinothiazine (**DPT-2**, with two quinoline rings) were obtained according to previously described methods [20,21]. The 1 mM stock of both phenothiazines was prepared in 100 % DMSO (Sigma Aldrich, Poznań, Poland), and before addition to the cells, the appropriate solutions, in fresh, complete sterile DMEM-F12, were prepared (final solutions used for biological experiments were as follows: 100, 50, 25, 12.5; 6.25; 3.15; 1.56; 0.78 µM). DMEM-F12 medium, trypsin, sodium phosphate buffer saline (PBS, pH = 7.4), and doxorubicin were bought from Merck (Poznań, Poland). Annexin-V apoptosis assay was obtained from BioLegend (San Diego, CA, USA). Propidium iodide solution (PI) was obtained from BD Biosciences (San Jose, CA, USA). Fetal bovine serum (FBS, EURx, Gdańsk, Poland), physiological saline (PBS without Ca and Mg, PAN-Biotech Gmbh, Aidenbach, Germany), and Annexin-V binding buffer (BD Biosciences, San Jose, CA, USA) were used after dissolving sterile H_2_O 10 times prior to usage [23].

### 3.2. Cell Culturing

Biological experiments were conducted on a panel of cancer cell lines, adenocarcinoma human alveolar basal epithelial cells (A549), and human non-small cell lung carcinoma cell line (H1299). In order to determine the potential side effects of treatments, research was performed simultaneously on the human bronchial epithelial cells (BEAS-2B) and normal human dermal fibroblasts (NHDF); all cells received were received from the ATCC collections (Manassas, VA, USA). Cells were grown in DMEM F-12 medium (PAA, Warsaw, Poland) supplemented with 10% (*v*/*v*) heat-inactivated FBS (Eurx, Gdańsk, Poland) and 100 units/mL penicillin and 100 μg/mL streptomycin (Sigma-Aldrich, Darmstadt, Germany) at 37 °C in a humidified atmosphere with 5% CO_2_ [23,27,29].

### 3.3. MTT Viability Assay

Examination of cytotoxicity using MTT assay is the most convenient and fastest method for screening of cytotoxic action of the drugs. MTT assay is a colorimetric method for evaluation of cell viability based on mitochondrial dehydrogenase enzyme activity. The survival rate of the cells was calculated relative to the control cells, grown in standard conditions. Using this assay, IC_50_ values were calculated for each cell line treated with standard compounds and drugs. For MTS assay, cells were seeded in 96-well plates at 1 × 10^4^ cells/well in 0.2 mL completed medium 24 h before drug treatment. In the next step, cells were incubated for 72 h with the compounds (doses: 100; 50; 25; 12.5; 6.25; 3.15 and 1.56 µM) and then were washed three times with PBS (PAA, Laboratories GmbH, Cölbe, Germany). They were then incubated for 2 h with 20 μL of MTT solution (0.5 mg/mL; Promega) in 100 μL of PBS (PAA, Laboratories GmbH, Cölbe, Germany) until the color in the control changed from light yellow to purple, and the colorimetric reaction was developed. Finally, the formazan crystals were dissolved in 75 μL of isopropanol/HCl mixture (*v/v* 1:0.04). The levels of absorbance were measured at a wavelength of λ = 570 nm using a microplate spectrophotometer (Epoch; BioTek, Winooski, VT, USA) and expressed as a % of the untreated control, named the survival fraction [SF] [23,27,29].

### 3.4. Cell Cycle Analysis and Apoptosis/Necrosis Analysis via Flow Cytometry Using Annexin-V Assay

Many drugs work by inhibiting cell proliferation and causing cell cycle blockage. Using flow cytometry, the impact of new compounds on the cell cycle were determined. Cells incubated with compound were also analyzed for cellular death pathways, apoptosis induction, followed by Annexin-V assay. Annexin-V is a protein that binds phosphatidylserine and allows early detection of apoptosis; additionally, iodium propide staining distinguishes it from necrosis. For apoptosis and cell cycle assays the cell cultures were plated in 6-well plates at a confluence of 1 × 10^5^ in 2 mL of completed fresh DMEM-F12 medium, 24 h before drugs addition. An Aria III flow cytometer (Becton Dickinson; Franklin Lakes, NJ, USA) was used. The flow cytometry analysis was done on a free Flowing Software 2.5.1 program (by Perttu Terho, the Cell Imaging and Cytometry Core, Turku Bioscience Centre, Turku, Finland, with the support of Biocenter Finland) [23,27,29].

### 3.5. Microscopic Observation

Microscopic observation and images acquisition with Live Cell Analyzer (JuLI™ Br; NanoEnTek Inc., Seoul, Korea) was used to directly observe on-plate confluence, density, and cell viability and images acquisitions [30,31].

### 3.6. Statistical Analyses

The results are expressed as means ± SD from three independent experiments. Results were analyzed in MS Excel 2010. Statistical significance was calculated with a *t*-test, and a *p*-value < 0.05 is indicated with a star [23,24,25,26,27,28,29,30,31].

### 3.7. ADME Analysis and Target Prediction

In silico analyses of the molecular descriptor and parameters of Lipinski’s, Ghose’s and Veber’s rules and an ADME profile were carried out using a Swiss internet server SwissADME [32]. Prediction of biological targets was carried out using web platform Way2Drug [33].

## 4. Conclusions

In this publication, we presented two synthetic diazaphenothiazines, tricyclic 10*H*-3,6-diazaphenothiazine (**DPT-1**, with two pyridine rings) and pentacyclic 7-(3′-dimethylaminopropyl)diquinothiazine (**DPT-2**, with two quinoline rings), demonstrating significant in vitro anticancer activity against lung cancer cell lines, the human lung carcinoma A549 and non-small lung carcinoma H1299, and protective potential to healthy cell lines, BEAS-2B and NHDF. The reference compound in the conducted studies was doxorubicin. Using a 72 h MTT viability assay (Promega) we observed a strong cytotoxic activity of **DPT-1** and **DPT-2**.

Both compounds appear to be good candidates for in vitro tests, especially in 3D cultures, where a mixture of sensitive healthy cell lines is used together with chemoresistant cancer cell lines. A low lethal effect observed in NHDF or BEAS-2B cells at IC_50_ doses against A549 or H1299 cells confirms promising selectivity against cancer. The analysis of the cell cycle revealed different pathways of the mechanism of anticancer activity induced by the studied diazaphenothiazines. **DPT-1** activated the process of apoptosis, in contrast to **DPT-2**, which activated the necrosis phase. Both tested compounds showed selectivity of action. Preliminary in silico analyses of ADME parameters and the biological profile confirmed the validity of our research.

Based on these primary findings, it can be concluded that **DPT-1** and **DPT-2** possess potential as chemotherapeutic agents. Further advanced in vivo and enzymatic studies are planned.

## Figures and Tables

**Figure 1 molecules-27-05255-f001:**
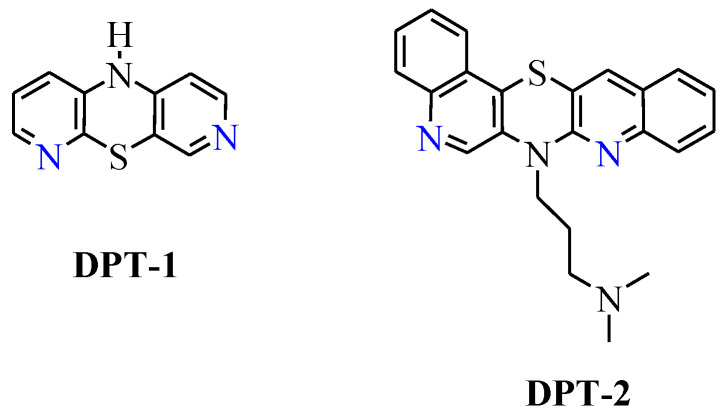
Structures of diazaphenothiazines **DPT-1** (10*H*-3,6-diazaphenothiazine) and **DPT-2** (7-(3′-dimethylaminopropyl)diquinothiazine).

**Figure 2 molecules-27-05255-f002:**
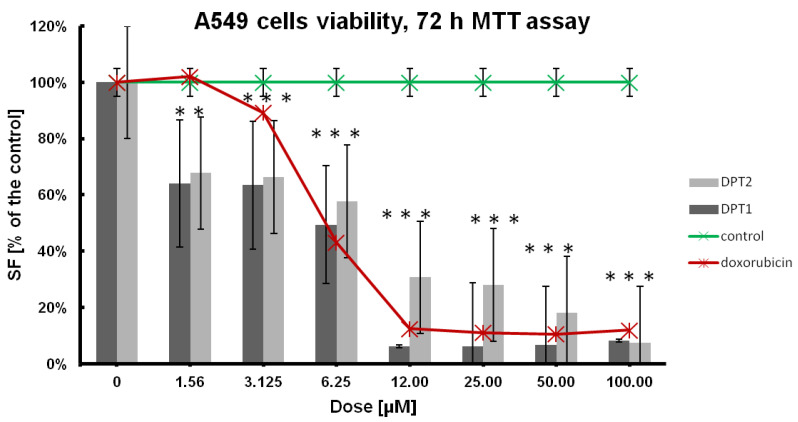
Dose effect of the **DPT-1** and **DPT-2** treatment on the viability of A549 cells after 72 h of treatment, followed by MTT assay (control—untreated cells; positive control—doxorubicin-treated cells). Data are presented as mean ± SD. *** or **—statistical significance in comparison to the control was calculated with a *t*-test, and *p*-value < 0.05 is indicated with a star.

**Figure 3 molecules-27-05255-f003:**
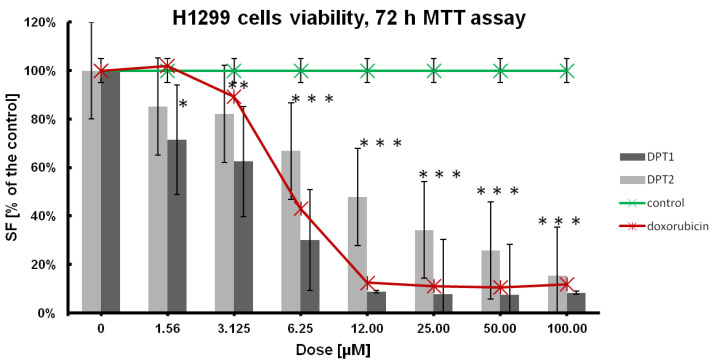
Dose effect of the **DPT-1** and **DPT-2** treatment on the viability of H1299 cells after 72 h of treatment, followed by MTT assay (control—untreated cells; positive control—doxorubicin-treated cells). Data are presented as mean ± SD. *** or ** or *—statistical significance in comparison to the control was calculated with a *t*-test, and *p*-value < 0.05 is indicated with a star.

**Figure 4 molecules-27-05255-f004:**
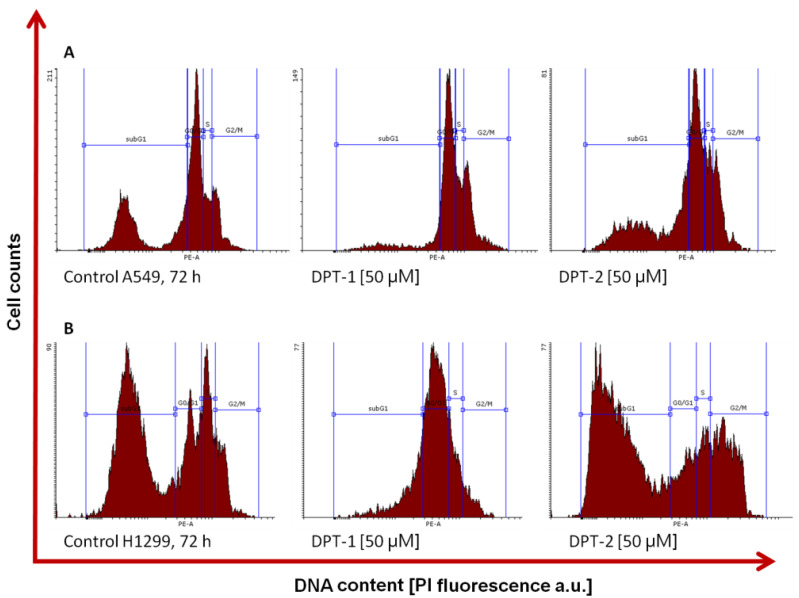
Typical histograms of control and **DPT-1**- and **DPT-2**-treated A549 (**A**) and H1299 cells (**B**) treated for 72 h at a dose of 50 µM, followed by flow cytometry cell cycle measurements.

**Figure 5 molecules-27-05255-f005:**
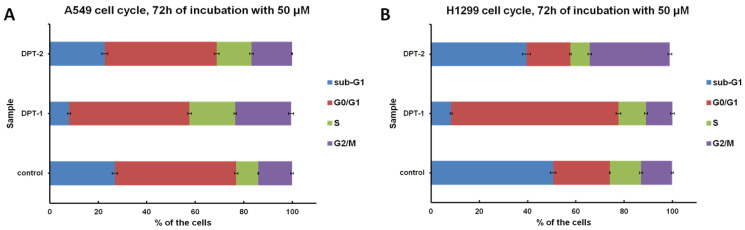
Cell cycle distribution in control and **DPT-1**- and **DPT-2**-treated A549 (**A**) and H1299 cells (**B**) treated for 72 h at a dose of 50 µM, followed by flow cytometry measurements.

**Figure 7 molecules-27-05255-f007:**
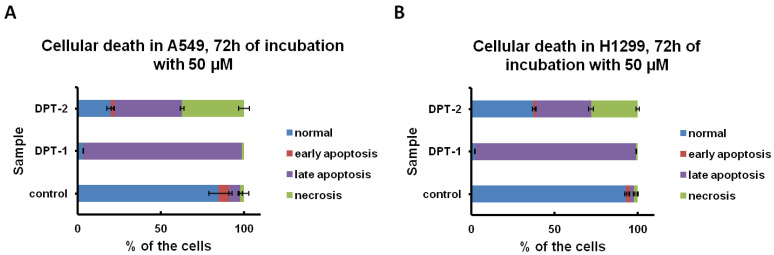
Cellular death in control and **DPT-1**- and **DPT-2**-treated A549 (**A**) and H1299 cells (**B**) treated for 72 h at a dose of 50 µM, followed by flow cytometry measurements of Annexin-V apoptosis assay and iodium propide (PI) staining (normal cells: Annexin-V_−_/PI_−_; early apoptosis: Annexin-V_+_/PI_−_; late apoptosis: Annexin-V_+_/PI_+_; necrosis: Annexin-V_−_/PI_+_).

**Figure 8 molecules-27-05255-f008:**
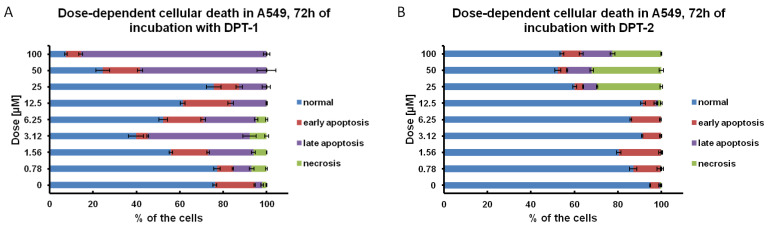
Dose-dependent cellular death in control and (**A**) **DPT-1**- and (**B**) **DPT-2**-treated A549 cells treated for 72 h, followed by flow cytometry measurements of Annexin-V apoptosis assay and iodium propide (PI) staining (normal cells: Annexin-V_−_/PI_−_; early apoptosis: Annexin-V+/PI_−_; late apoptosis: Annexin-V_+_/PI_+_; necrosis: Annexin-V_−_/PI_+_).

**Figure 9 molecules-27-05255-f009:**
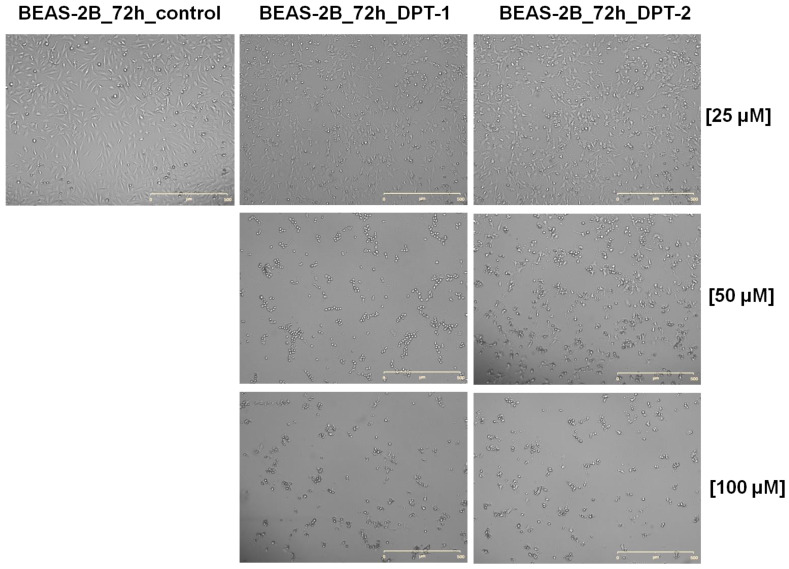
Dose-dependent cellular death in control and **DPT-1**- and **DPT-2**-treated BEAS-2B cells treated for 72 h, followed by Live Cell Analyzer images acquisitions. Magnification 100×; scale bar 500 µM.

**Figure 10 molecules-27-05255-f010:**
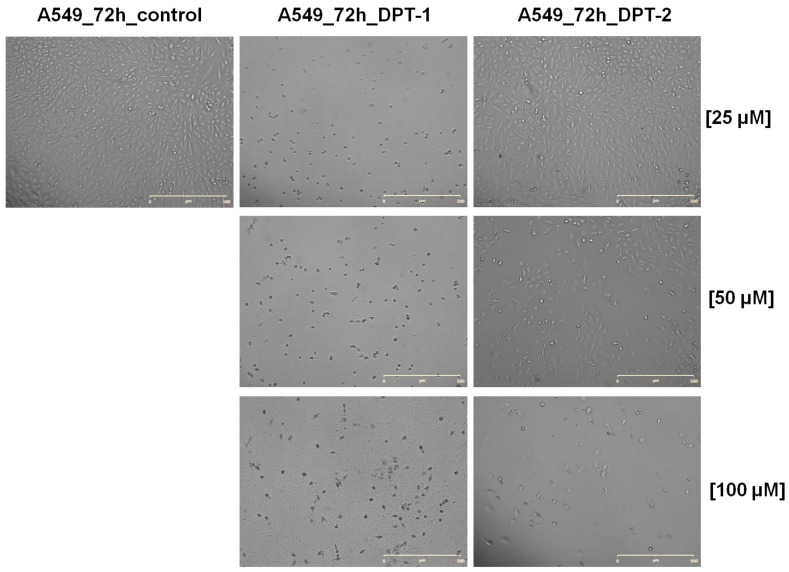
Dose-dependent cellular death in control and **DPT-1**- and **DPT-2**-treated A549 cells treated for 72 h, followed by Live Cell Analyzer images acquisitions. Magnification 100×; scale bar 500 µM.

**Table 1 molecules-27-05255-t001:** IC_50_ value after 72 h of incubation with tested phenothiazines and doxorubicin for tissue-dependent effects in NHDF, A549, and H1299 cell lines.

IC_50_ [µM]	Cell Line			
	NHDF	BEAS-2B	A549	H1299
**DPT-1**	2.246 ± 0.01	2.4679 ± 0.01	1.526 ± 0.004	2.515 ± 0.005
**DPT-2**	18.77 ± 0.038	11.2648 ± 0.038	3.447 ± 0.054	12.895 ± 0.013
doxorubicin	116.061 ± 0.002	0.0651 ± 0.001	5.289 ± 0.004	7.749 ± 0.004

Mean values from three experiments ± standard deviation.

**Table 2 molecules-27-05255-t002:** The molecular descriptor and parameters of Lipinski’s, Ghose’s and Veber’s rules for **DPT-1** and **DPT-2 [32]**.

No	Molecular Mass (M)	H-bond Acceptors	H-bond Donors	Rotatable Bonds	TPSA	Lipinski’s Rules	Ghose’s Rules	Veber’s Rules
**DPT-1**	201	2	1	0	63	+	+	+
**DPT-2**	386	3	0	4	57	+	+	+

**Table 3 molecules-27-05255-t003:** The ADME activities predicted for **DPT-1** and **DPT-2** [32].

No	LogP_calc._	GI Absorption	BBB Permeant	Pgp Substrate	CYP1A2 Inhibitor	CYP2C9 Inhibitor	CYP2D6 Inhibitor
**DPT-1**	1.78	high	+	+	+	+	+
**DPT-2**	4.45	high	+	+	+	+	+

**Table 4 molecules-27-05255-t004:** Probability of activities of **DPT-1** and **DPT-2** using PASS Program [33].

DPT-1 Probability of Activity	DPT-2 Probability of Activity
Histone deacetylase SIRT1 stimulant Histone deacetylase stimulant Angiogenesis factor	X-methyl-His dipeptidase inhibitor Tumor necrosis factor alpha release inhibitor Immunomodulator
Anti-inflammatory	Angiogenesis factor

## Data Availability

The datasets used and analyzed during the current study are available from the corresponding authors on reasonable request.

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
