# Peer review of "Tri- and Pentacyclic Azaphenothiazine as Pro-Apoptotic Agents in Lung Carcinoma with a Protective Potential to Healthy Cell Lines"

_molecules, 2022, doi:10.3390/molecules27165255_

Round 1
Reviewer 1 Report
The authors examined the anticancer effects of the two compounds using MTT, cell cycle and flow cytometry. Even though the activity part of the study was well planned, I think that the activity studies of the two existing compounds are insufficient in terms of medicinal chemistry. Therefore, I think that the study is not suitable especially for this section of the journal.
- The authors obtained 2 compounds by previously described method, but did not present any evidence that they performed structure determinations of these compounds. The structures of the compounds need to be determined by mass spectroscopy, NMR. In addition, purity checks should be done by HPLC.
- In the activity parts of the compounds, the authors performed MTT and flow cytometry, but an experimental procedure was not used to determine how the compounds showed this activity.
- Enzymatic pathways can be tried or in silico studies can be performed.
- It is also impossible to discuss structure-activity, since the number of compounds is only two.
Although it is a good study in terms of activity results, I do not think it is a suitable medicinal chemistry article to be published in this journal. After the authors increase the number of items and complete the analysis procedures, they can provide resubmission with a broader and more enlightened activity pathway. However, it is not suitable for publication as it is.
Author Response
The response to the reviewer 1
Thank you for all comments and suggestions:
Comments and Suggestions for Authors
The authors examined the anticancer effects of the two compounds using MTT, cell cycle and flow cytometry. Even though the activity part of the study was well planned, I think that the activity studies of the two existing compounds are insufficient in terms of medicinal chemistry. Therefore, I think that the study is not suitable especially for this section of the journal.
- The authors obtained 2 compounds by previously described method, but did not present any evidence that they performed structure determinations of these compounds. The structures of the compounds need to be determined by mass spectroscopy, NMR. In addition, purity checks should be done by HPLC.
- In the activity parts of the compounds, the authors performed MTT and flow cytometry, but an experimental procedure was not used to determine how the compounds showed this activity.
- Enzymatic pathways can be tried or in silico studies can be performed.
- It is also impossible to discuss structure-activity, since the number of compounds is only two.
- Although it is a good study in terms of activity results, I do not think it is a suitable medicinal chemistry article to be published in this journal. After the authors increase the number of items and complete the analysis procedures, they can provide resubmission with a broader and more enlightened activity pathway. However, it is not suitable for publication as it is.
- The authors obtained 2 compounds by previously described method, but did not present any evidence that they performed structure determinations of these compounds. The structures of the compounds need to be determined by mass spectroscopy, NMR. In addition, purity checks should be done by HPLC.
Answer: Full structural characteristics of both compounds have been presented in the publications:
- Morak‐Młodawska, B.; Pluta, K.; Latocha, M.; Suwińska, K.; Jeleń, M.; Kuśmierz, D. 3,6‐Diazaphenothiazines as potential lead molecules—Synthesis, characterization and anticancer activity. J. Enzym. Inhib. Med. Chem. 2016, 31, 1512–1519. doi:10.3109/14756366.2016.1151014 (Ref. 20)
- Jeleń M.; Pluta K.; Latocha M.; Morak-Młodawska B.; Suwińska K.; Kuśmierz D. Evaluation of angularly condensed diqui-nothiazines as potential anticancer agents. Bioorg. Chem. 2019, 87, 810-820. DOI: 10.1016/j.bioorg.2019.04.005 (Ref.21),
and it includes: NMR, 2D NMR (COSY, ROESY, HMBC, HSQC) analyzes and HR MS (High Resolution Mass Spectrometry) by ESI technique. The presented biological experiments are a continuation and did not require the publication of previously published data confirming the structure and purity of the compounds.
- In the activity parts of the compounds, the authors performed MTT and flow cytometry, but an experimental procedure was not used to determine how the compounds showed this activity.
Answer: We do not agree with the Reviewer’s comment, a standard procedure for determination of agent’s anticancer activity and mechanism of its action is tested followed by viability assays (MTS, MTT, SRB etc.). Based on the literature, previous reports and newly obtained results the IC50 value (inhibitory concentration reduced cells viability in 50%, in comparisons to the untreated control) of the for anti-proliferative and toxic agents were assessed. Additionally we have present the mode of action using a cell cycle analysis, followed by flow cytometry. We have discussed a cytostatic action of tested compounds, very carefully and we have showed a potential of phenothiazines as a perfect blockers of the cell cycle. The Reviewer omitted the Annexin-V apoptosis assay, where we eventually confirmed the anticancer potential of tested compounds and we have showed their pro-apoptotic and pro-necrotic potential. Finally, the microscopic evaluation fulfill the determination of the anticancer activity. In our opinion, the biological evaluation is completely, well discussed and confirmed by the presented in manuscript results or previously presented reports from the literature.
- Enzymatic pathways can be tried or in silico studies can be performed.
Answer: Further enzymatic tests as well as advanced in vivo tests are planned in the next stage of the research.
- It is also impossible to discuss structure-activity, since the number of compounds is only two.
Answer: The studied compounds were identified as leading structures in previous studies (DPT-1 was selected from among 14 dipyridothiazines and DPT-2 from 23 diquinothiazines, Ref. 20 and 21). The presented studies are an extension of previous screening studies for the most active derivatives. Of course, we agree with the fact that two compounds are not enough to consider the structure-activity relationship, and therefore in our work we did not deal with it but focused on the high anticancer potential of the studied derivatives.
- Although it is a good study in terms of activity results, I do not think it is a suitable medicinal chemistry article to be published in this journal. After the authors increase the number of items and complete the analysis procedures, they can provide resubmission with a broader and more enlightened activity pathway. However, it is not suitable for publication as it is.
Answer: In response to the review, we would like to explain that the manuscript was written at the invitation of the editor, which was sent in the first edition: “Design, Synthesis, and Analysis of Potential Drugs “ for Dr. Anna Kasprzycka. The topic has been accepted. Unfortunately, due to pandemic reasons, the manuscript was prepared with a delay and was sent to the second edition of a special issue “Design, Synthesis, and Analysis of Potential Drugs, 2nd Edition“ (also with a renewed invitation to the editor).
In response to the review, we would like to thank you for evaluating our manuscript. Changes are marked in yellow.

Reviewer 2 Report
The manuscript entitled “Tri- and pentacyclic azphenothiazine as a pro-apoptotic agents 2 in lung carcinoma with a protective potential to healthy cell 3 lines” reported good study, however the present study is just the extension of previously reported study. Herein, authors reported numerous studies in different cancer cell lines as compared to previously reported cell lines which seems to just extend without significant. However apoptosis studies are interesting which make the paper to consider in the Molecules. Following are concern regarding manuscript before set decision:
1. In Fig 2. : For DPT2: the % of the control is almost same at 12 and 25 µM, but then decreases for 50 and 100 µM. How? Explain this.
2. In Fig 2. : For DPT1: the % of the control remains same for 12 and 25 µM, but then increases for 50 and 100 µM. How? Explain this.
3. Authors should explain PDT1 and PDT2.
4. In flow cytometry analysis (Figure 4) and Cell cycle distribution (Figure 5) authors perform study at 50 µM. Why? Explain this.
5. Why authors selected BEAS-2B cells irrespective of H1299 cells in Microscopic long-term live observations? Explain this.
Author Response
The response to the reviewer 2
Thank you for all comments and suggestions:
- In Fig 2. : For DPT2: the % of the control is almost same at 12 and 25 µM, but then decreases for 50 and 100 µM. How? Explain this.
Answer: The saturation of the receptors, responsive to the cellular death in tested cell lines occurred at doses higher than 6.25 µM, what means that a maximal cytotoxic effect will be presented above presented in Fig. 2 border dose of 6.25 µM. Incubation of cells with any higher doses, resulted with similar, nearly 100 % of viability reduction effects – that’s why higher doses 12 and 25 µM did not induce linear effects – we have reached almost plateau in cell killing.
- In Fig 2. : For DPT1: the % of the control remains same for 12 and 25 µM, but then increases for 50 and 100 µM. How? Explain this.
Answer: In physiology and pharmacology such effect is called a bimodal effect, often connected to the hormesis effect (lower doses displayed better effects than higher – sensitisation of the receptors is over, and any higher doses did not resulted with better toxicity any more). However, any effects for the DPT-1 or DPT-2 presented in Fig. 2 at doses range 12-100 µM did not differ (please see the standard deviation bars). The toxic effect are reached above 6.25 µM for both compounds and seemed to be maximal.
- Authors should explain PDT1 and PDT2.
Answer: The chemical names of used in the studies chemicals are well described in first sentence of the abstract and in the introduction sections: phenothiazine derivatives, tricyclic 10H-3,6-diazaphenothiazine (DPT-1) and pentacyclic 7-(3`-dimethylaminopropyl)diquinothiazine (DPT-2).
- In flow cytometry analysis (Figure 4) and Cell cycle distribution (Figure 5) authors perform study at 50 µM. Why? Explain this.
Answer: For better evaluation and comparison to the common used anticancer drugs, such as doxorubicin, we have choose for presentation of Figure 4 ad 5 in cell cycle distribution a dose with 10 times higher concentration, than for doxorubicin’s IC50 , reached in the A549 cell line (please see the Table 1). In our opinion, the effects similar between 6.25-100 µM observed by MTT viability assay, could be tested using one of doses in further assays – we have chosen 50 µM, the living cells were still observed during long-term effects (please see images from Figure 9 and 10).
- Why authors selected BEAS-2B cells irrespective of H1299 cells in Microscopic long-term live observations? Explain this.
Answer: The aim of the studies presented in manuscript, was to show a strong anticancer potential of phenothazines derivatives with slightly lower toxicity against healthy cells. BEAS-2B cells were chosen for comparison of effects, using a live long-term observations, and the images showed typically morphological changes after expositions to the drugs. The cells BEAS-2B seemed to be still in good condition after dose of 25 µM, whereas cancer cell line starts to die (please compare the Figures 9 and 10).
In response to the review, we would like to thank you for evaluating our manuscript and for your favorable review. Changes are marked in yellow.

Reviewer 3 Report
Comments:
1. Page 3, line 112, Page 10, line 270, and Page 11, line 295, “range: 100; 50; 25; 12.5; 6.25; 3.15; 1.56 μM.” should be amended “range: 100, 50, 25, 12.5, 6.25, 3.15, 1.56 μM.”
2. Page 3, in the “Table 1”, The IC50 of doxorubicin: 52.899 ± 0.004 is different with in the “Figure 2”, should be 5.2899 ± 0.004 ? The authors should reconfirm this results and Standard deviation in the “Table 1”.
3. Page 5, line 174, “DPT-1 and DPT-2” should be amended “DPT-1 and DPT-2”.
4. Page 6, in the “Figure 4” and “Figure 6”, in the cell cycle assay, in the control group, the three phases of the cells are very incomplete in the two cell lines, and the Sub-G1 phase is very large, indicating that the cells are in an unhealthy state, and the authors should re-analyze this experiment of cell cycle assay and apoptosis assay.
5. Page 10, line 265, “7-(3′-dimethylaminopropyl)diquino[3,2-b;3′,4-e]thiazine” should be amended “7-(3′-dimethylaminopropyl)diquinothiazine”.
6. Page 11, line 327, “10H-326 3,6-diazaphenothiazine (dipyrido[2,3‐b;4’,3’‐e][1,4]thiazines) (DPT-1, with two pyridine rings) and pentacyclic 7-(3′-dimethylaminopropyl)diquino[3,2-b;3′,4′-e]thiazine (DPT-2, with two quinoline rings)” should be amended 10H-326 3,6-diazaphenothiazine (dipyrido[2,3‐b;4’,3’‐e][1,4]thiazines) (DPT-1, with two pyridine rings) and pentacyclic 7-(3′-dimethylaminopropyl)diquinothiazine (DPT-2, with two quinoline rings)”.
7. Page 12, line 378, “Clin. Ches.t Med. 2011, ” should be amended “Clin. Chest. Med. 2011, ”.
8. Page 13, line 394, “J.Mol.Struct. 2020, 1216, 1-28.” should be amended “J. Mol. Struct. 2020, 1216, 1‒28.”.
9. Page 13, line 439, “J. Inorganic Biochem. 2019” should be amended “J. Inorg. Biochem. 2019”.
10. In the “References”: The format and arrangement in the “References” should conform to Molecules.
Author Response
The response to the reviewer 3
Thank you for all comments and suggestions:
- Page 3, line 112, Page 10, line 270, and Page 11, line 295, “range: 100; 50; 25; 12.5; 6.25; 3.15; 1.56 μM.” should be amended “range: 100, 50, 25, 12.5, 6.25, 3.15, 1.56 μM.”
Answer: We have changed that.
- Page 3, in the “Table 1”, The IC50 of doxorubicin: 52.899 ± 0.004 is different with in the “Figure 2”, should be 2899 ± 0.004 ? The authors should reconfirm this results and Standard deviation in the “Table 1”.
Answer: Yes, because of our mistake, the comma were in the wrong place, the IC50 for doxorubicin is calculated for A549 cell line, and it should be 5.2899 ± 0.004. We have change that in Table 1 and corrected in text.
- Page 5, line 174, “DPT-1 and DPT-2” should be amended “DPT-1 and DPT-2”.
Answer: We have corrected the mistake.
- Page 6, in the “Figure 4” and “Figure 6”, in the cell cycle assay, in the control group, the three phases of the cells are very incomplete in the two cell lines, and the Sub-G1 phase is very large, indicating that the cells are in an unhealthy state, and the authors should re-analyze this experiment of cell cycle assay and apoptosis assay.
Answer: We do agree, that the control’s sub-G1 phase is high, but not extremely: it is circa 25% and 50% for A549 and H1299 cell respectively (please, see the Figure 5). This is a typically situation for the high proliferative cell lines, it is also the effect of long-term incubations. It will be unbelievable it the sub-G1 phase would not be presented on the Figures for control, we would like to show really situation, with the physiological consequences also in control panels (please, confirm with the control images from Figures 9 and 10). All the, so called background effects weren’t hidden by us.
- Page 10, line 265, “7-(3′-dimethylaminopropyl)diquino[3,2-b;3′,4-e]thiazine” should be amended “7-(3′-dimethylaminopropyl)diquinothiazine”.
Answer: We have corrected the name.
- Page 11, line 327, “10H-326 3,6-diazaphenothiazine (dipyrido[2,3‐b;4’,3’‐e][1,4]thiazines) (DPT-1, with two pyridine rings) and pentacyclic 7-(3′-dimethylaminopropyl)diquino[3,2-b;3′,4′-e]thiazine (DPT-2, with two quinoline rings)” should be amended 10H-326 3,6-diazaphenothiazine (dipyrido[2,3‐b;4’,3’‐e][1,4]thiazines) (DPT-1, with two pyridine rings) and pentacyclic 7-(3′-dimethylaminopropyl)diquinothiazine (DPT-2, with two quinoline rings)”.
Answer: We have corrected the names.
- Page 12, line 378, “Clin. Ches.t Med. 2011, ” should be amended “Clin. Chest. Med. 2011, ”.
Answer: We have corrected. - Page 13, line 394, “J.Mol.Struct. 2020, 1216, 1-28.” should be amended “J. Mol. Struct. 2020, 1216, 1‒28.”.
Answer: We have corrected. - Page 13, line 439, “J. Inorganic Biochem. 2019” should be amended “J. Inorg. Biochem. 2019”.
Answer: We have corrected. - In the “References”: The format and arrangement in the “References” should conform to Molecules.
Answer: We have corrected
In response to the review, we would like to thank you for evaluating our manuscript and for your favorable review. Changes are marked in yellow.

Round 2
Reviewer 3 Report
No comment!
Author Response
The response to the reviewer 3
Thank you for all comments and suggestions. Following the editor's instructions, the in silico research section has been added and the discussion has been expanded. In response to the review, we would like to thank you for evaluating our manuscript and for your favorable review. Changes are marked in yellow.
